# Beneficial Effects of the Five Isolates of *Funneliformis mosseae* on the Tomato Plants Were Not Related to Their Evolutionary Distances of SSU rDNA or PT1 Sequences in the Nutrition Solution Production

**DOI:** 10.3390/plants10091948

**Published:** 2021-09-18

**Authors:** Jingyu Feng, Zhe Huang, Yongbin Zhang, Wenjing Rui, Xihong Lei, Zhifang Li

**Affiliations:** 1Beijing Key Laboratory of Growth and Developmental Regulation for Protected Vegetable Crops, Department of Vegetable Science, College of Horticulture, China Agricultural University, Haidian District, Yuanmingyuanxilu 2, Beijing 100193, China; mirror0406@163.com (J.F.); huangzhe@cau.edu.cn (Z.H.); zyb950326@163.com (Y.Z.); ruiwenjin@163.com (W.R.); 2Beijing Agricultural Extention Station, Huixinxili 10, Changyang District, Beijing 100029, China; leixihong@126.com

**Keywords:** arbuscular mycorrhizal fungi, soilless culture, nutrient uptake, tomato yield, soil beneficial microorganisms

## Abstract

The symbiosis and beneficial effects of arbuscular mycorrhizal fungi (AM fungi) on plants have been widely reported; however, the effects might be unascertained in tomato industry production with coconut coir due to the nutrition solution supply, or alternatively with isolate-specific. Five isolates of AM fungi were collected from soils of differing geographical origins, identified as *Funneliformis mosseae* and evidenced closing evolutionary distances with the covering of the small subunit (SSU) rDNA regions and Pi transporter gene (PT1) sequences. The effects of these isolates on the colonization rates, plant growth, yield, and nutrition uptake were analyzed in tomato nutrition solution production with growing seasons of spring–summer and autumn–winter. Our result indicated that with isolate-specific effects, irrespective of geographical or the SSU rDNA and PT1 sequences evolution distance, two isolates (A2 and NYN1) had the most yield benefits for plants of both growing seasons, one (E2) had weaker effects and the remaining two (A2 and T6) had varied seasonal-specific effects. Inoculation with effective isolates induced significant increases of 29.0–38.0% (isolate X5, T6) and 34.6–36.5% (isolate NYN1, T6) in the plant tissues respective nitrogen and phosphorus content; the plant biomass increased by 18.4–25.4% (isolate T6, NYN1), and yields increased by 8.8–12.0% (isolate NYN1, A2) compared with uninoculated plants. The maximum root biomass increased by 28.3% (isolate T6) and 55.1% (isolate E2) in the autumn–winter and spring–summer growing seasons, respectively. This strong effect on root biomass was even more significant in an industry culture with a small volume of substrate per plant. Our results reveal the potential benefits of using selected effective isolates as a renewable resource that can overcome the suppressing effects of sufficient nutrient availability on colonization rates, while increasing the yields of industrially produced tomatoes in nutrition solution with coconut coir.

## 1. Introduction

Arbuscular mycorrhizal fungi (AM fungi) are naturally soil-borne benefit microorganisms of terrestrial plants, acting as renewable biological resources. Over 80% of plants can establish this symbiotic partnership, which is essential for the uptake of inorganic phosphate (P), nitrogen (N) and various trace elements, while preventing or reducing abiotic and biotic stresses on plants [1,2,3].Tomato (*Solanum lycopersicum* L.), which is the most widespread vegetable species consumed globally [4], is known to have symbiotic partnerships with AM fungi, which promote plant growth and development [5]. For example, inoculation with AM fungi (*Rhizophagus. intraradices)* was reported to increase the P content of tomato fruits by 60% [6]. Industrial soilless cultivation of fresh market tomatoes has become popular because of the improved growth, yields, and quality of these commodities grown under such conditions [7]. In these cultures, the root system of the plants develops within a small amount of growing medium, the nutrient supply must be adequate, and the root system must be efficient at taking up water and nutrients [8]. Therefore, the use of AM fungi in industrial tomato production may potentially strengthen the plant’s root system and the efficiency of nutrient uptake. However, some studies have reported that AM fungi have no effect on plant growth and the uptake of nutrients from a solution in the case of tomatoes grown in sand or peat as soilless culture [9].

This discrepancy was attributed to a lower rate of colonization in hydroponic cultures that is likely to be caused by the presence of ample P concentrations in the root zones of plants in a nutritional solution culture [8,10]. Potted tomato plants inoculated with AM fungi (*Funneliformis mosseae* and *R. intraradices*) showed higher marketable fresh yields, mainly at lower levels of fertilization (half- and quarter-strength nutrient solutions) [11,12]. Azcoin et al. found that AM fungus inoculation resulted in higher N uptake from fertilization in the presence of medium concentrations of N (6 mM) [13], whereas the reverse trend was observed with high amounts of N fertilization (9 mM) [14]. Similarly, P uptake, growth, and biomass were only found to increase in mycorrhizal plants in cases of low P availability [8,15], while high soil concentrations of P or fertilizer doses could reduce mycorrhizal abundance [16,17]. All of these findings seemingly indicate that the beneficial effects on the plants nutritional uptake and growth are strongly dependent on the availability of P and N. From an economic perspective, the efficiency of industrial tomato production is ensured by high yields and the use of technologies that can easily be adapted to hydroponic systems [18]. Balanced and adequate nutrient availability is one of the main requirements for ensuring high fruit yields of tomato plants. However, some studies have found that plants colonized by AM fungi produce higher yields compared with the yields from uninoculated plants, irrespective of the use of hydroponic systems [19]. Therefore, further research and a deeper understanding of symbiosis are required to overcome the bottleneck of AM fungi utilization in industrial tomato production.

According to Rouphael et al., (2015), AM fungus inoculation has particular effects associated with variations in genotypes and environmental conditions [12]. The dynamic responses of plants to both the interacting partners and the environmental conditions merit consideration [20,21]. There is evidence that ‘functional diversity’ relating to plant and fungal genotypes determines the benefits resulting from these interactions, with some combinations being more effective than others in terms of nutrition and/or stress resistance [21,22,23]. For example, one study found that colonization of plant roots by *F. mosseae* and *R. irregularis*, which are the most prevalent AM fungi, resulted in different levels of disease control and drought tolerance [24]. The effectiveness of the colonization process was found to be contingent on the cultivar or ecotype and the mycorrhizal species, although there was no strict specificity of symbiotic partners [25]. Mycorrhizal plants show different patterns of adaptation to environmental stresses depending on the fungus species [26]. Similarly, maximal multiple benefits were obtained using AM fungi isolates that were found to be effective after ensuring the accurate selection of compatible species/genotype-fungus combinations and the adoption of favorable management practices [12]. Accordingly, we hypothesized that AM fungi isolates with different origins or evolutionary distances of SSU rDNA or PT1 sequences have different functional effects on tomato plants in spite of receiving sufficient nutrition, including N and P supplies. This study was aimed at determining the impacts of mycorrhizal association with the effects of five isolates of *F. mosseae* collected from geographically distant sites on plants of two growing seasons under industrial production using coconut coir. Specifically, we examined the effectiveness of root colonization as well as plant growth and yield responses, N and P uptake, and root biomass.

## 2. Material and Methods

### 2.1. Experimental Site and Design

#### 2.1.1. Isolation and Identification of the AM Fungi

The following five AM fungi isolates were collected from wild soils in different geographical locations: (1) the isolate NYN1 (109°57′ E and 39°19′ N), (2) the isolate A2 (116°10′ E and 40°08′ N), (3) the isolate T6 (129°57′ E and N46°42′ N), (4) the isolate X5 (129°46′ E and 46°54′ N), and (5) the isolate E2 (10°59′ E and 47°25′ N). They were collected using the wet-sieving method [27]. One gram of soil from each of these sites was added into a pot planting white clover (*Trifolium repens* L.) with sterilized sands around 3 months. Afterwards, the spores in each pot were collected with wet-sieving, 5 spores according to same morphology of *F. mosseae* were collected and propagated with white clover again in sands until enough spores were produced. The species and their genetic relationships were identified on the basis of both morphology and the gene SSU (covers the small subunit SSU) rDNA regions [28]; and the Pi transporter gene (PT1) which was identified as phosphate transporter gene, a gene marker for the identification and discrimination of AM fungi in the Genus *Glomus* [29]. Polymerase chain reaction (PCR) analysis of the SSU and PT1 with specific primers for AM fungi and sequencing was performed, as described below.

#### 2.1.2. DNA Extraction, Amplification and Sequencing

Spores were extracted through 30% sucrose with 4000 density gradient centrifugation, and wet sieving was used for DNA extraction [30]. Approximately 40–60 spores from each isolate were placed in an Eppendorf tube. A sterile steel ball was inserted into the sample tube, and the sample was crushed using a plant tissue crusher (45 kHz, 30 s). DNA extraction was performed using the cetyltrimethylammonium bromide (CTAB) method (the DNA kit was provided by Aidlab Biotechnology Co. Ltd. Beijing, China). A fragment of approximately 1500 bp, covering the entire SSU rDNA regions was amplified using the nested protocol. In the first PCR, a reaction mix of 50 µL was prepared using primers GeoA1 (5′-GGTTGATCCTGCCAGTAGTC-3′) and ART4 (5′-TCCGCAGGTTCACCTACGG-3′). The nested PCR reaction were performed by diluting the first PCR amplification (1:100) and using the primers GeoA2 (5′-CCAGTAGTCATATGCTTGTCTC-3′) and Geo11 (5′-ACC TTGTTACGACTTTTACTTCC-3′) [31]. To enable further identification of the relationships among the different isolates, we used the Pi transporter gene (PT1) region to reveal genetic differences among isolates [29], and the sequence identified by [32]. The following primer pairs were designed using Primer Premier V6.0 (PREMIER Biosoft International, 3786 Corina Way, Palo Alto, Canada 94303-4504): (1) Pm1-F (5′-TTCCGCCATGGGTACTGTTC-3′) and Pm1-R (5′-CGAAATGCCATGACCGGTTG-3′) and (2) Pm2-F (5′-CGTCTCACGGTCCCAGAATC-3′) and Pm2-R (5′-TCGGGAAGACTTCTCCTGGT-3′).

#### 2.1.3. Sequence and Phylogenetic Analysis

The PCR products for DNA sequencing analysis were conducted by the company (Tsingke Biological Technology Co., Ltd., Beijing, China). The results of sequences (GenBank Banlit submission. Submission ID: **2441966**) were compared with those shown on the website of the National Center for Biotechnology Information (https://www.ncbi.nlm.nih.gov/; accessed on 26 April 2021). Accordingly, we selected the ten closest sequences and aligned them with the Molecular Evolutionary Genetics Analysis across Computing Platforms version 5.0 MEGA X software (Mega Limited, Auckland, New Zealand) (https://www.megasoftware.net; accessed on 26 April 2021). The systematics of the *Glomeromycota* [33,34] all neighbor-joining (NJ) phylogenetic analyses were computed using the MEGA X software, with 1000 bootstraps. Referring to Kumar et al. (2018) [35], we performed a molecular evolutionary genetics analysis, viewing and editing phylogenetic trees with the MEGA X software.

### 2.2. Experimental Site and Design

#### 2.2.1. Experimental Protocol and Growing Seasons

The experiments were performed from September 2018 to March 2019, covering the autumn–winter growing season (tomato cultivar, ‘Futesi’) and from April to July 2019, covering the spring–summer growing season (cultivar, ‘Qidali’), with coconut coir used as the growth medium. The experimental site was a greenhouse located at the Agricultural Technology Extension Centre in Beijing, China (40°10′ N and 116°24′ E). The ‘Qidali’ tomato cultivar (with big fruit, F1) was suitable for spring–summer season and obtained from Beijing Golden World Seeds Co., Ltd., Beijing, China; and the ‘Futesi’ cultivar (small fruit, F1) was suitable for autumn–winter season and obtained from Rijk Zwaan Distribution B.V., De Lier, Netherlands.

#### 2.2.2. AM Fungi Material and Inoculation

The plants inoculated with the five AM fungi isolates were compared with the uninoculated control plants. After transplanting, each plant for the AM fungi treatments, there was 30 g of AM fungus inoculant with approximately 50 spores per gram and well mixed with the growth media of coconut coir moss. The same amount of inoculant washed microorganisms and nutrients were added to each control pot for the non-AM fungi treatments. The microorganism wash was prepared as follows: the inoculant sands were drained with distilled water to collect possible microorganisms; then, the drainage was filtered through filter paper (Hangzhou tezhong zhiye GmbH, Hangzhou, China) with a particle retention potential of 5–8 mm. The washed sands were sterilized afterwards. Both the filtered solution and the residues sterilize sand was added to the non-AM fungi treatments, to keep the same amount of nutrition and other microorganisms from the inoculant in non-AM fungi treatments [27].

#### 2.2.3. Growth Conditions of Tomato Plants

Seven-week tomato seedlings were transplanted on 5 September 2018 (the autumn–winter growing season) and on 15 March 2019 (the spring–summer season), with an average density of 2.4 plants per m^2^. The growth medium, coconut coir moss, is commercially available and is imported from Sri Lanka (by Shouguang Lvtian International Trade Co., Ltd. Shouguang, China). This substrate is commonly used in soilless industrial tomato production in China. A split plot design with four replicates was applied with each treatment covering a surface area of 58.3 m^2^. Uniform seedlings, cultivated using the same method, were randomly transplanted in the plots. The plants were grown in a solar-powered greenhouse, with day and night temperatures of 25 ± 4 °C and 16 ± 4 °C, respectively and humidity between 60–80%, under natural radiation. The plants were fertilized with a nutrient solution modified following Hoagland and Arnon (1950) that supported the growth of tomato plants [36]. The substrates (per liter) comprised 165 mg of N and 100 mg Ca added as KNO_3_ and Ca (NO_3_)_2_∙4H_2_O, 55 mg of P as KH_2_PO_4_, 220 mg of K and 65 mg S as K_2_SO_4_, 50 mg of Mg as MgSO_4_∙7H_2_O, 10.4 mg of Fe as Fe-EDTA, 10 mg of Zn as ZnSO_4_∙7H_2_O, and 10 mg of Cu as CuSO_4_∙5H_2_O based on dry substrates. The average nutritional solution supply was 1200–1600 mL per plant per day according to the plants’ growth requirements.

### 2.3. An Analysis of Biomass, Yields, Nutrients, and Colonization Rates

Five plants from each subplot were randomly sampled, and 20 plants were assessed per treatment at the end of the 5-month growing season. The samples of shoots were oven dried for 48 h at 70 °C for the biomass and nutrient analysis. The roots were carefully removed from the growth medium, washed, weighed to record their fresh weight and separated into two sections. One section was dried to determine the biomass and nutrients, and the other section was soaked in 30% alcohol to analyze the AM fungi colonization rate. The N and P concentrations in the shoots were measured using a DC plasma ccelle spectrometer (Beckman Instruments, Beijing, China) and the Kjeldahl digestion method. The root colonization rate was determined using the method described by Koske and Gemma [37]. The yield was recorded and summed according to each harvest.

### 2.4. Statistical Analysis

The data were recorded on MS Excel sheets and analyzed using the IBM SPSS software to determine mean values and standard errors. The statistical results derived from the experiment were expressed as means ± SE. The differences among the means were analyzed via a one-way ANOVA, followed by Fisher’s least significant difference *t* test (LSD) for multiple comparisons test (*p* ≤ 0.05) to determine if significant differences existed among the plants inoculated with AM fungi isolates and the uninoculated control. A univariate analysis of variance was also performed to analyze the main effects observed for the AM fungi isolates and the control sample. We have not compared the statistical differences of the data between two growing seasons due to different cultivars; one was big fruit another was small fruit.

## 3. Results

### 3.1. Isolate Identification and Their Root Colonization

After the DNA extraction was completed, PCR with specific primers for AM fungi and sequencing were conducted to verify the AM fungi identities and variability. Although the isolates were from different geographical areas, both the SSU rDNA and PT1 region sequences of the *Glomerales* (Pm1 and Pm2 sequences) revealed the closeness of the isolates E2 and NYN1, and T6 and A2 (Figure 1A–C). The frequency of root fragment colonization by hyphae of mycorrhizal in tomato plants inoculated with the different isolates evidenced significant differences compared with uninoculated plants at the end of cultivation (Figure 2). The effectiveness of the inoculation was found to vary according to the isolates and growing seasons. The isolate A2 evidenced the highest colonization rate, followed by the T6 and NYN1 during the autumn–winter growing season (‘Futesi’); NYN1 had the highest colonization rate, followed by A2 and T6, and lastly X5 and E2 during the spring–summer season (‘Qidali ’) (Figure 2).

### 3.2. Effects of Mycorrhizae on Plant Growth and Yields

There were significant differences in plant biomass and yields observed between the inoculated and control plants and among all five isolates during the two growing seasons (Table 1). During the autumn–winter season (for the ‘Futesi’ cultivar), the biomass of the shoots of the plants colonized by the isolates T6, NYN1, and X5 increased by 20.7%, 22.0% and 20.1%, respectively, compared with those of the control plants. Further, the biomass of roots in plants that were colonized by the isolates T6, NYN1, and A2 increased by 28.3%, 12.8% and 21.2%, respectively. The yields increased the most when the isolates NYN1, A2, and X5 were added, with these isolates inducing increases in the total biomass by 25.4%, 18.9%, and 16.8%, respectively, in this season (Table 1).

During the spring–summer season (for the ‘Qidali’ cultivar), inoculation of plants with the isolates NYN1, T6, and A2 increased the yields by approximately 8.8%, 6.9% and 6.7%, respectively, and the total biomass by 16.5%, 18.4%, and 11.3%, respectively, compared with those of the control plants. Further, the biomass of the shoots evidenced a significant increase of 25.2% when the T6 was introduced compared with that of the shoots in the control plants, while the root biomass showed significant increases following inoculation with the E2, NYN1, and T6 by 55.1%, 50.4%, and 47.6%, respectively (Table 1).

### 3.3. Effects of Mycorrhizae on Plant Nutrient Uptake

Colonization by these five isolates significantly increased N and P uptakes of tomato plants compared with those of the uninoculated plants in both seasons (Table 2 and Table 3). During the autumn–winter season of the ‘Futesi’ cultivar, among the experimental isolates, only the X5 significantly influenced N concentrations in the shoot tissue (8.1%), while the T6 and A2 significantly influenced N concentrations (28.7% and 14.4%) in the root tissue of the plants. During this season (‘Futesi’ cultivar), the isolates A2, T6, X5, and NYN1 stimulated greater N accumulation in the shoots per plant compared with those of the uninoculated plants, with increases of 23.0%, 23.0%, 31.1%, and 26.2%, respectively. And N accumulation in the root tissue was higher for the A2 and T6, with increases of 37.4% and 64.2%, respectively (Table 3).

During the spring–summer season (the ‘Qidali’ cultivar), the isolates A2, T6, X5, and NYN1 increased N concentrations in the shoots of plants by 8.9%, 10.5%, 12.7%, and 9.3%, respectively, compared with uninoculated plants (25.9 mg g^−1^). Further, N concentrations in the root tissue were affected by the treatments A2 and T6 induced increased N concentrations in the root tissue of plants (7.0% and 6.2%, respectively) during this season (Table 2). While the T6 and NYN1 induced higher N accumulation in the plant shoot tissue (increases of 38.2% and 31.1%, respectively), and the E2, T6, and NYN1 induced higher N accumulation in the root tissue, with increases of 56.0%, 57.9% and 54.2%, respectively, compared with those of uninoculated plants during this season (the ‘Qidali’ cultivar) (Table 3).

The isolates T6, X5, and NYN1 affected P concentrations in the shoot tissue of plants, which increased by 7.9%, 9.2% and 11.8%, respectively, compared with those of the uninoculated plants in autumn–winter season (the ‘Futesi’ cultivar). The isolates A2, T6, and X5 induced significant increases in P concentrations in the root tissues (24.7%, 30.4%, and 14.7%, respectively) compared with those of the uninoculated plants in this season (Table 2). Further, the P accumulation in the tomato shoots was influenced by the isolates A2, T6, X5, and NYN1, with increases of 21.6%, 29.7%, 31.5% and 36.3%, respectively; the A2, T6, and X5 induced higher P accumulation in the root tissue, with increases of 50.8%, 67.6%, and 27.1%, respectively, in this season (Table 3).

During the spring–summer season (the ‘Qidali’ cultivar), the isolate T6 increased P concentrations in the shoots of plants (by 8.0%) compared with the uninoculated plants whereas P concentrations in the root tissue were not affected by inoculations in any of the five isolates (Table 2). In this season, the P accumulation in the shoots of tomato plants was induced by the T6 and NYN1 isolates, with increases of 35.5% and 25.3%, respectively, compared with that of uninoculated plants; P accumulation the root tissue were higher in plants inoculated with the E2 and NYN1, with increases of 73.8% and 80.8%, respectively, compared with those of the non-inoculated plants (Table 3).

## 4. Discussion

### 4.1. Mycorrhizal Colonization of Roots

In the present study, in the presence of an adequate nutrient supply, the colonization rates of the five tested isolates of *F. mosseae* in the roots of inoculated tomato plants were significantly increased compared with the uninoculated plants, but with isolate-specific (Figure 2). AM fungi colonization was found to vary in plants grown in soilless media [8,38], with colonization associated with cultivation on perlite, peat, coir, and sawdust [8,10,19,39,40]. In the present study, the isolates E2 and A2, respectively, evidenced the lowest and highest root colonization rates for tomato plants at 32% and 52% during the autumn–winter growing season; the isolates E2 and NYN1 were associated with the lowest and highest rates of root colonization at 38% and 55%, respectively, during the spring–summer season (Figure 2). The mycorrhizal frequency for the uninoculated plant roots was less than 7% during both growing seasons (Figure 2). This small amount of colonization was also reported by the other study that it is possibly because of the air-borne transfer of propagules from the inoculated plants [8]. From our study, within the 5-month growing seasons under field conditions, it was possible with the air-borne transfer of spores. Nevertheless, this small amount of colonization was not taken a substantial effect on the results.

These values of colonization rates are lower than those obtained by Maboko et al. [10], who reported root colonization rates of 78.2% for tomato plants grown in coconut and 78% for those grown in sawdust. However, another study conducted by Dasgan et al. found that the colonization rate in both open (without recirculation) and closed (with recirculation) fertigation systems was approximately 28% [19], while Cwala et al. found that the colonization rate ranged between 14% and 25% in hydroponic cultivation [40]. Kowalska et al. found that the colonization rates of tomato roots were affected by both the growing times (days after inoculation) and the specific P level of the nutrient solution [8]. Overall, low levels of root colonization were attributed to an abundant supply of nutrients, especially P, available to the plants [41]. In a study entailing the same P level as that used in the present study, Kowalska et al. reported a 33% colonization rate in plants grown in a coconut culture [8]. The differences in colonization levels in the present study, in which nutrient supplies were adequate, can be attributed to the diversity of the isolates and their interactions with the growing seasons (Figure 2). The rates of post-inoculation root colonization by hyphae of mycorrhizal in the plants during both growing seasons were 52% and 54%, respectively, for the isolates A2 and NYN1, and 32% for the E2. Furthermore, A2 and T6 achieved stable colonization that exceeded 50% for plants during both growing seasons, while colonization rates associated with X5 and NYN1 varied according to the season (and cultivar), and E2 demonstrated a weak ability to achieve stable colonization (Figure 2). Another study also found that colonization rates varied for different species: *F. mosseae* (56%) and *Claroideoglom. etunicatum* (63%) [42] (Rafique et al., 2018). Although all of the tested isolates in the present study were from the same species, *F. mosseae*, there were nevertheless significant differences among the different isolates (Figure 2). Therefore, colonization is evidently influenced by different isolates regardless of the level of nutrients supplied. In this context, the effect of nutrients on AM fungi symbiosis might be attenuated by selecting fungi isolates or species of effective symbiosis applied in soilless industrial tomato production entailing nutrient solution supplies.

### 4.2. Biomass and Yield Production

As presented in Table 1, in the present study, successful colonization of five isolates of AM fungi induced higher biomass and yields for the inoculated tomato plants compared with the uninoculated plants. Variability in growth and yields was also found among the different isolates (Table 1). During the autumn–winter growing season, the shoot biomass of the mycorrhizal plants inoculated with the T6, X5, and NYN1 increased by 20.7%, 20.1%, and 22.0%, respectively, and evidenced similar colonization levels. However, these effects were not produced by the A2. The root biomass of the plants significantly increased following inoculation with the A2, T6, and NYN1—but not with the X5—by 21.2%, 28.3%, and 12.8%, respectively (Figure 2). In this context, the influences of the AM fungi isolates on biomass accumulation did not always correspond with the shoot and root N and P content or with the colonization rate. The yields and total biomass also varied, with most increases associated with the isolates NYN1, A2, and X5. The yields increased by 9.1%, 12.0%, and 11.0%, respectively, using these isolates, and the total biomass increased by 25.4%, 18.9%, and 16.8%, respectively, compared with those of the control plants. However, the plants inoculated with T6 did not evidence increased yields and total biomass, differing from root biomass in autumn-winter season (Table 1). It was still an open question that this root biomass increasing was not extending to shoot biomass and yield accumulation of T6 treatment. Further study could be concerning this point.

During the spring–summer season, the yields of tomato plants inoculated with the isolates NYN1, T6, and A2 increased by approximately 8.8%, 6.9%, and 6.7%, respectively, while their total biomass increased by 16.5%, 18.4%, and 11.3%, respectively, compared with those of the control plants (Table 1). In this season, the A2 was associated with the highest yields and root biomass but not with the highest shoot biomass (Table 1). In addition, T6 induced a significant increase in the dry biomass of the shoots by 25.2% relative to the control, while root biomass increased significantly by 55.1%, 50.4% and 47.6%, respectively, following inoculation with the E2, NYN1, and T6 (Table 1).

In sum, the A2 and NYN1 had stable positive effects on the growth and yields of tomato plants of both growing seasons. While the beneficial effect of X5 was more pronounced in the autumn–winter season, while that of T6 was more apparent in the spring–summer season. Significantly higher yields—as much as 9.1% and 12.0% in autumn-winter season, and 8.8% and 6.7% in the spring–summer season occurred after inoculation with NYN1 and A2, respectively (Table 1).

The tested isolates induced increases with a range of 6.7–12.0% in the yields of the two growing seasons in this soilless environment, using coconut coir as the growth medium. This increase was less than the 23% figure reported by Affokpon et al. [43], who inoculated tomato plants in an open field with a mixture of AM fungi isolates. In general, the soil water and plant nutrition status in open fields are not controlled as well as they are under greenhouse conditions with soilless production. Maboko et al. found that under greenhouse conditions of soilless production, the AM fungi (four arbuscular mycorrhizal species, *Glomus etunicatum, Paraglomus occultum, Glomus clarum,* and *Glomus mosseae*) have limited effects on tomato production [10]. However, another study found that under conditions of severe, moderate, and mild drought stress, the AM fungi *R. intraradices* affected yields, which increased by 25%, 23%, and 16%, respectively, compared with those of uninoculated plants [44]. The results of the present study revealed increased yields of approximately 6.7–12.0% in response to selected isolates of AM fungi under conditions of adequate nutrient supply, indicating the economic value of applying these isolates in tomato production systems using coconut coir as the growth medium.

The isolate E2 evidenced the lowest colonization level and had no significant effect on the biomass growth of tomato plants of either growing seasons (Table 1). Although this isolate induced an increase in root biomass, this positive effect did not impact on shoot growth or yields (Table 1). The yield response was also correlated with the higher colonization rate of A2, but not with the colonization rate of NYN1 compared with the other four isolates (Figure 2). Maboko et al. found that despite relatively high root colonization rates of AM fungi in tomato plants (78.2% and 77.7% in coir and sawdust, respectively) [10], there was no significant improvement in tomato yields, indicating that a higher colonization rate is only one of the factors—and not the deciding one—that influences the effectiveness of AM fungi. These varying results could be attributable to different nutrient supplies, especially N and P availability, and the different AM fungi species or isolates used in the different studies.

The results of the present study showed that the responses of root biomass to the effective AM fungi isolates were more pronounced for plants of the spring–summer season compared with those of the autumn–winter season. However, this relatively higher root biomass did not correspond to higher total biomass or yields (Table 1). The significant influence of these AM fungi isolates on root biomass remains strong even in soilless culture because soilless cultures are designed to foster well-developed root systems because of the small substrate volumes available per plant, as the absorption surface of the roots via the presence of AM fungi could significantly benefit the plants [45]. Because one of the benefits of successful colonization of plants in soilless cultures is strongly developed root systems, the plants could absorb large quantities of water and nutrients as a result of the increased absorption surface of the roots via the AM fungi association [8].

### 4.3. Nitrogen and Phosphorus Uptake

Many previous studies have reported that the application of mycorrhization results in remarkable improvements in nutrition status of plants, especially their N and P nutrition status [46]. Variability in nutritional dynamics observed in the present study was caused by differences among the isolates in growing seasons (Table 2). In the autumn–winter season, N concentrations in the shoot tissue were only influenced by the isolate X5, which did not have the same effect in the roots, whereas T6 induced significantly increased N concentrations in the root tissue but not in the shoots (Table 2). T6, X5 and NYN1 affected P concentrations in the shoot tissue, while the A2, T5, and X5 significantly increased P concentrations in the root tissue during the autumn–winter growing season (Table 2). In the spring–summer season, the isolates A2, T6, X5, and NYN1 increased N concentrations in the shoots, whereas only the A2 and T6 increased N concentrations in the root tissue (Table 2). Moreover, only the T6 increased P concentrations in the shoots, with none of the isolates affecting P concentrations in the root tissue of the plants during this season (Table 2). In sum, the N and P concentrations in the shoot and root tissues of the plants were not always inherent to the holistic mycorrhizal plants. This finding may imply that the different isolates affect the distribution of these two main plant nutrients. The effects of the AM fungus inoculations were found to be greater on N concentrations in plants during in the spring–summer growing season than during the autumn–winter growing season, whereas the reverse was true for P concentrations, which were greater for plants during the autumn–winter season (Table 2).

Because of the interaction effects of nutrient concentrations and plant biomass accumulation, the N and P content in the mycorrhizal plant tissue were significantly higher compared with those of the uninoculated plants. Moreover, these levels varied among the isolates and between growing seasons (Table 3). Higher accumulation of both N and P in the tomato plant shoots were associated with the isolates A2, T6, X5, and NYN1, whereas higher N content levels in the root tissue were stimulated by the A2 and T6 during the autumn–winter season. Further, the isolates A2, T6, and X5 induced higher P content levels in roots of inoculated plants compared with these levels in the uninoculated plants (Table 3). Higher N and P accumulation in the shoot tissue of plants were stimulated by T6 and NYN1, while higher N accumulation in the root tissue were stimulated by E2, T6 and NYN1 during spring–summer. And the isolates E2 and NYN1 showed higher P accumulations in the root tissue compared with those of the uninoculated plants in this season (Table 3). In sum, the total content of N and P in the plants in both seasons were most influenced by T6 and NYN1, with E2 evidencing positive effects on N and P accumulation in spring-summer season, and X5 showing in the autumn–winter season (Table 3). There were corresponding increases in the biomass, growth, and yields of inoculated plants with the isolates (Table 1 and Table 3). Thus, the effects of the different isolates on N and P accumulation were varied but correlated with the growing seasons (together with cultivars) used in this study. Considering the plant biomass accumulation and nutrient uptake together, we found the same pattern of responses of selected effective isolates on both plant growth and nutrient accumulation (Table 1 and Table 3).

Labour et al. observed that studies of mycorrhizae in natural ecosystems and low tillage agriculture clearly illustrate the multiple benefits of this ancestral symbiosis in any climate and geographical location [47]. Moreover, they suggested that inoculation and the subsequent management of AM fungi could lead to advances in agricultural practices but would require more knowledge concerning the functioning of the two symbiotic partners. In the present study, the *F. mosseae* isolates from far distances in China and Germany, although with same the species identification according to morphology and ITS, and near evolution based on PT1 sequence, their influence on the growth of tomato plants were varied in growing seasons with sufficient nutrient supplies in soilless industrial cultures. The yields of the inoculated plants were correlated with higher nutritional status among the isolates. The higher yields were induced by increasing biomass accumulation, which was a result of nutritional benefits from inoculation with effective isolates of *F. mosseae*.

## 5. Conclusions

The results of this study revealed that the fruit yields and mineral status of tomato plants responded to specific *F. mosseae* isolates in symbiotic partnership produced in a coconut coir growth medium. The effective isolates were associated with high colonization rates in spite of the adequate supply of nutrients. Although with the closing evolutionary distances based on both SSU rDNA and PT1 region sequences of these five isolates, two of them had positive effects of yield on plants of both the growing seasons, two varied in their influence on yield of both seasons/cultivars and one had a weaker influence relative to the uninoculated plants. Inoculation with more effective isolates was found to stimulate N and P uptake in the plant tissues, which increased significantly by 29.0–38.0% and 34.6–36.5%, respectively, while the biomass and fruit yields increased by 18.4–25.4% and 8.8–12.0%, respectively. In conclusion, this study is the first to compare the influence of *F. mosseae* isolates collected from soils obtained from different geographical locations that were used as renewable biological resources on plant yields and allows more effective use of chemical fertilizer in soilless industrial tomato production.

## Figures and Tables

**Figure 1 plants-10-01948-f001:**
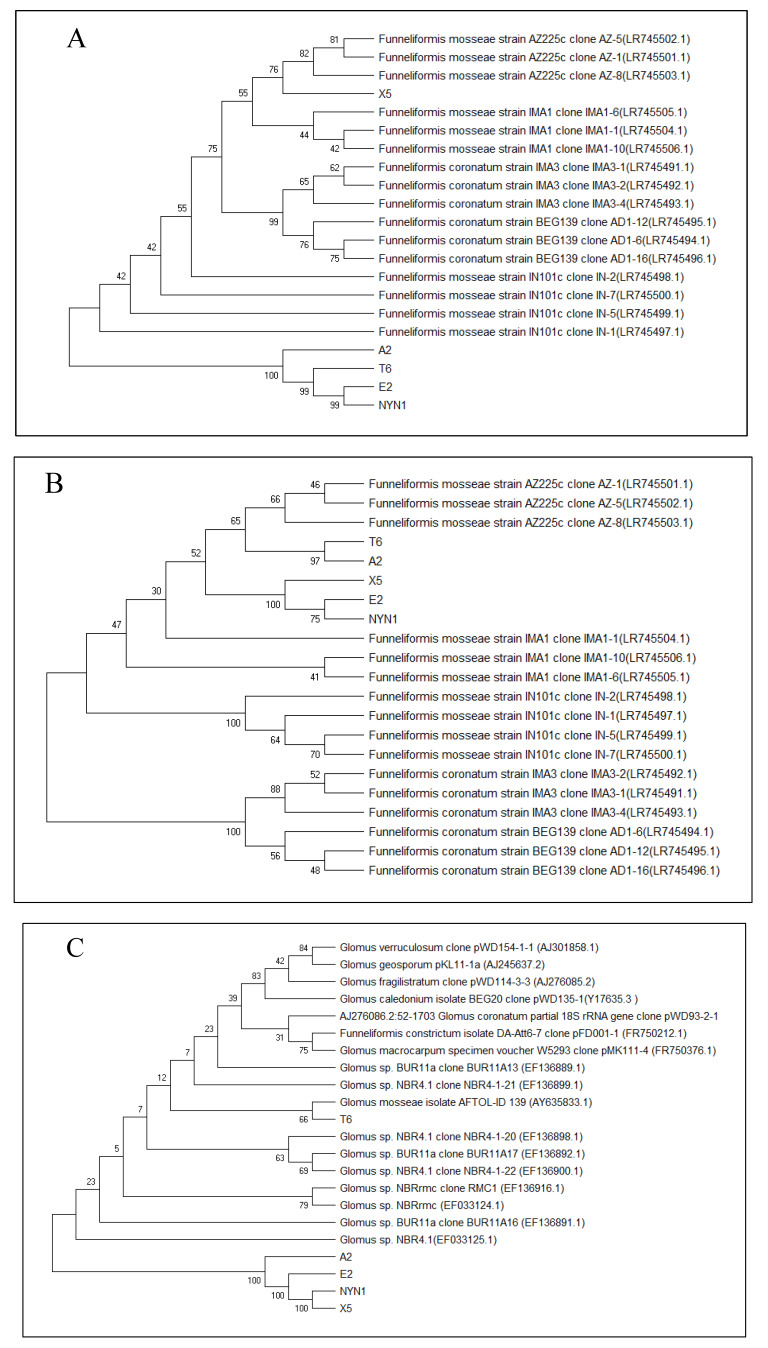
Pm1 sequences (**A**), Pm2 sequences (**B**) and AM fungi isolates identified on the basis of both PT1 and SSU rDNA (**C**) region sequences of the *Glomerales*. The percentage of replicate trees in which the associated taxa clustered together in the bootstrap test (1000 replicates) are shown next to the branches. The tree is drawn to scale, with branch lengths in the same units as those of the evolutionary distances used to infer the phylogenetic tree. The evolutionary distances were computed using the maximum composite likelihood method and are in the units of the number of base substitutions per site. Evolutionary analyses were conducted in MEGA X. (**A**) The length of partial sequence of PT1 gene is 320 bp, and the sum of branch length = 1.21661574; (**B**) The length of partial sequence of PT1 gene is about 680 bp, and the sum of branch length = 0.22664112. (**C**) The length of partial sequence of SSU fragment is about 1600 bp, and the sum of branch length = 126719649.

**Figure 2 plants-10-01948-f002:**
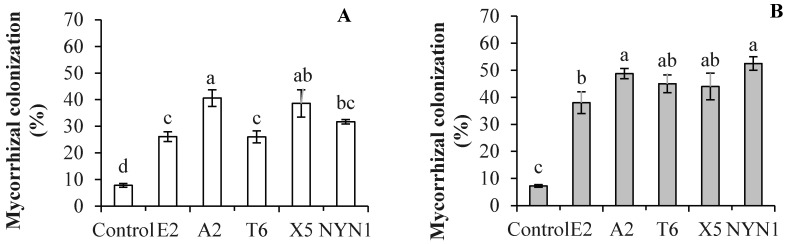
A comparison of tomato root colonization (%) by isolates of arbuscular mycorrhizal fungi during the autumn–winter growing season (**A**) and the spring–summer season (**B**). Different letters indicate significant differences among isolates as revealed in a one-way ANOVA and followed by a Fisher’s least significant difference *t* test (LSD test) at *p* ≤ 0.05.

**Table 1 plants-10-01948-t001:** Effects of AM fungi on biomass and yield of tomato plants of two growing seasons.

Treatment	Shoot Biomass (g plant^−1^)	Root Biomass (g plant^−1^)	Fruit Biomass (g plant^−1^)	Total Biomass (g plant^−1^)	Yield (kg m^−2^)
Autumn–Winter (cv. ‘Fortesa’)					
Control	47.2 ± 1.69bc	5.15 ± 0.15d	45.8 ± 0.71b	98.2 ± 2.02c	3.09 ± 0.08b
Isolate E2	46.7 ± 2.81c	5.55 ± 0.21cd	49.9 ± 3.73ab	102.2 ± 6.21bc	3.31 ± 0.15ab
Isolate A2	55.3 ± 3.33ab	6.24 ± 0.34ab	55.2 ± 1.78ab	116.8 ± 3.28a	3.46 ± 0.05a
Isolate T6	56.7 ± 2.31a	6.61 ± 0.17a	46.8 ± 2.70b	110.2 ± 1.46abc	3.21 ± 0.05ab
Isolate X5	57.0 ± 4.03a	5.70 ± 0.23bcd	52.0 ± 2.32ab	114.7 ± 3.53ab	3.43 ± 0.11a
Isolate NYN1	57.6 ± 1.14a	5.81 ± 0.07bc	59.7 ± 8.47a	123.1 ± 8.24a	3.37 ± 0.06a
Spring–Summer (cv.‘Qidali’)					
Control	84.9 ± 5.12b	4.68 ± 0.34b	159.9 ± 4.08b	249.5 ± 3.43b	13.3 ± 0.08c
Isolate E2	100.7 ± 9.30ab	7.26 ± 1.45a	162.2 ± 15.0ab	270.1 ± 16.4ab	13.6 ± 0.15bc
Isolate A2	90.4 ± 5.93ab	5.23 ± 0.30ab	182.0 ± 3.26a	277.7 ± 3.99a	14.2 ± 0.05ab
Isolate T6	106.3 ± 3.92a	6.91 ± 0.23a	182.2 ± 3.05a	295.4 ± 4.65a	14.2 ± 0.05ab
Isolate X5	90.9 ± 7.04ab	6.00 ± 0.86ab	177.0 ± 3.73ab	273.9 ± 9.82ab	14.0 ± 0.11abc
Isolate NYN1	102.2 ± 6.08ab	7.04 ± 0.26a	181.4 ± 1.40a	290.6 ± 5.58a	14.5 ± 0.06a

Fruit biomass was identified after dried at 70 °C. Total biomass as the sum of dried shoot, root, and fruits. Yield is fresh weight of fruits. The values are expressed as the means ± SE. Different letters indicate significant differences (*p* ≤ 0.05, *n* = 4) among different treatments according to ANOVA and followed by Fisher’s least significant difference *t* test (LSD tests). Within each column, means followed by different letters are significantly different according to LSD test at *p* ≤ 0.05.

**Table 2 plants-10-01948-t002:** Effects of AM fungi on nitrogen and phosphorus concentrations in tomato plants of two growing seasons.

Treatment	N Concentration (mg g^−1^)		P Concentration (mg g^−1^)	
	Shoot	Root	Shoot	Root
Autumn–Winter(cv.‘Fortesa’)				
Control	25.9 ± 0.14b	16.7 ± 0.34bc	5.57 ± 0.06d	5.10 ± 0.24d
Isolate E2	26.1 ± 0.96ab	16.9 ± 0.77bc	5.74 ± 0.18cd	5.29 ± 0.08cd
Isolate A2	27.2 ± 0.62ab	19.1 ± 0.63a	5.78 ± 0.03bcd	6.36 ± 0.38ab
Isolate T6	26.4 ± 0.12ab	21.5 ± 1.61a	6.01 ± 0.06abc	6.65 ± 0.27a
Isolate X5	28.0 ± 0.40a	17.2 ± 0.57bc	6.08 ± 0.06ab	5.85 ± 0.16bc
Isolate NYN1	26.7 ± 1.09ab	16.4 ± 0.19c	6.23 ± 0.17a	5.33 ± 0.11cd
Spring–Summer(cv.‘Qidali’)				
Control	25.9 ± 0.48b	24.3 ± 1.18ab	7.34 ± 0.04b	5.55 ± 0.28a
Isolate E2	27.7 ± 0.87ab	24.7 ± 0.80ab	7.64 ± 0.12ab	6.17 ± 0.39a
Isolate A2	28.2 ± 0.97a	26.0 ± 0.57a	7.75 ± 0.21ab	5.99 ± 1.28a
Isolate T6	28.5 ± 0.62a	25.8 ± 0.54a	7.93 ± 0.23a	6.07 ± 0.48a
Isolate X5	29.2 ± 0.36a	22.4 ± 1.01b	7.78 ± 0.11ab	5.86 ± 0.47a
Isolate NYN1	28.3 ± 0.65a	24.8 ± 1.02ab	7.63 ± 0.08ab	6.68 ± 0.11a

The results are expressed as the means ± SE. Different letters indicate significant differences (*p* ≤ 0.05, *n* = 4) among different treatments according to ANOVA and followed by Fisher’s least significant difference *t* test (LSD tests). Within each column, means followed by different letters are significantly different according to LSD test at *p* ≤ 0.05.

**Table 3 plants-10-01948-t003:** The effects of AM fungi on nitrogen and phosphorus uptake in plants of two growing seasons.

	N Content			P Content		
	Shoot (g plant^−1^)	Root (mg plant^−1^)	Total (g plant^−1^)	Shoot (mg plant^−1^)	Root (mg plant^−1^)	Total (mg plant^−1^)
Autumn–Winter (cv. ‘Fortesa’)						
Control	1.22 ± 0.04b	86.2 ± 3.51c	1.31 ± 0.04b	263.0 ± 10.2b	26.2 ± 1.37c	289.2 ± 9.37b
Isolate E2	1.23 ± 0.12b	93.7 ± 4.43c	1.32 ± 0.11b	268.8 ± 21.1b	29.4 ± 1.51bc	298.2 ± 22.4b
Isolate A2	1.50 ± 0.08a	118.4 ± 3.23b	1.62 ± 0.08a	319.8 ± 20.2a	39.5 ± 2.42a	359.3 ± 20.3a
Isolate T6	1.50 ± 0.06a	141.5 ± 7.93a	1.64 ± 0.07a	341.0 ± 16.3a	43.9 ± 1.20a	384.9 ± 17.0a
Isolate X5	1.60 ± 0.11a	97.7 ± 3.40c	1.69 ± 0.11a	345.8 ± 21.1a	33.3 ± 1.23b	379.1 ± 21.8a
Isolate NYN1	1.54 ± 0.05a	95.4 ± 1.67c	1.63 ± 0.05a	358.4 ± 7.57a	30.9 ± 0.86bc	389.3 ± 8.05a
Spring–Summer (cv. ‘Qidali’)						
Control	2.20 ± 0.14b	113.0 ± 6.33b	2.31 ± 0.14b	623.3 ± 35.6c	26.0 ± 2.23b	649.2 ± 36.4c
Isolate E2	2.80 ± 0.31ab	176.3 ± 30.5a	2.98 ± 0.33ab	766.1 ± 59.7abc	45.2 ± 10.8a	811.3 ± 65.7ab
Isolate A2	2.57 ± 0.25ab	135.9 ± 5.27ab	2.71 ± 0.25ab	697.3 ± 31.2bc	31.5 ± 7.65ab	728.9 ± 28.6bc
Isolate T6	3.04 ± 0.17a	178.4 ± 8.11a	3.21 ± 0.18a	844.4 ± 49.2a	42.0 ± 3.89ab	886.4 ± 52.0a
Isolate X5	2.67 ± 0.24ab	136.5 ± 24.4ab	2.80 ± 0.26ab	707.1 ± 56.2abc	35.2 ± 5.49ab	742.3 ± 59abc
Isolate NYN1	2.89 ± 0.17a	174.2 ± 5.65a	3.07 ± 0.17a	781.0 ± 51.4ab	47.0 ± 1.05a	828.0 ± 50.8ab

The results of the analysis were expressed as mean ± SE values. Different letters indicate significant differences (*p* ≤ 0.05, *n* = 4) associated with different treatments according to the ANOVA and followed by Fisher’s least significant difference *t* test (LSD) test results. Within each column, the means followed by different letters are significantly different according to the results of LSD test at *p* ≤ 0.05.

## Data Availability

The data is contained within the article.

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
