# Peer review of "Beneficial Effects of the Five Isolates of Funneliformis mosseae on the Tomato Plants Were Not Related to Their Evolutionary Distances of SSU rDNA or PT1 Sequences in the Nutrition Solution Production"

_plants, 2021, doi:10.3390/plants10091948_

Round 1
Reviewer 1 Report
The manuscript entitled" Different beneficial efficiency of the strains of Funneliformis mosseae on the tomato plants not related to evolutionary distances of SSU rDNA or PT1 sequences in the nutrition solution production has interesting results. The authors have improved the version. The comments and quarries raised were answered well.
However, I suggest reframing the title again as the title of the article is not well suited to the article. Please reframe it again ---
Thanks for the revised version.
Author Response
Dear Madam/Sir,
Thank you for your comments. And we corrected as your suggested. The tittle changed as:
Beneficial effects of five isolations of Funneliformis mosseae on the plants were not related to their evolutionary distances of SSU rDNA or PT1 sequences in the tomato nutrition solution production
Best regards
Have a nice day.
Zhifang
Reviewer 2 Report
I have added my new comments to the Authors' replies. My new comments are introduced by ">", under the Authors' response. I still think that the manuscript is not ready for publication and it should be carefully revised.
Review 2:
Comments and Suggestions for Authors
Manuscript plants-1293422 deals with the effects of 5 different isolates of the arbuscular mycorrhizal fungus Funneliformis mosseae on two tomato cultivars, grown in greenhouse under semi-hydroponic conditions. The topic is consistent with the journal aims and scopes. English is always understandable, although it would greatly benefit from a native speaker revision. In general the manuscript could be considered for publication, but some issues, listed below, must be addressed by the Authors.
Language edited by Charlesworth Author Services: Order 78861.
> I am not satisfied with this. The text still contains several mistakes. For instance (and this is not meant to be a a complete list!): the sentence on lines 15, 16 and 17 must be rephrased, "affects" should be changed in "effects" and the rest of the sentence is not understandable (numbers referring to the pdf file I was sent by the publisher - I have noticed relevant differences in page numbers you refer into your replies and this file!); "the effects on colonization rate" - the effects of what?; "positive benefits" - benefits can only be positive! (line 23); Coconut COIR (not "coin"). So, please, have this checked carefully!
1. The title does not work. First, it is not correct grammatically, secondly it should be more synthetic and clear, thirdly it underlines the presence of 5 isolates (not strains) of F. mosseae, but it completely ignores that there are two plant cultivars. So, I recommend to find a better title.
We improved the title line 2-4.
> Again, the title is not satisfactory. Maybe: Beneficial effects of five different strains of...
2. The use of the word "strain" is common in microbiology, especially with reference to bacteria. For fungi, I am convinced that the word "isolate" is more appropriate. Please, modify throughout the manuscript.
We agree your suggestion, in the case that in fig 1, the reference species used “strain”. Our opinion is to keep this word. But it is still open for further discussing.
> I stick to my suggestion to use "isolate" and not "strain". You can change figure 1, this is not a valid reason to use strain.
3. The abstract is very poorly written and lacking a proper organization. It makes a sudden reference to "both growing seasons" but they were not mentioned before. Please try to summarize the paper: introduction, methods, results, comment.
We improved line 21-22.
> ok.
4. English: when you use a name with the function of an adjective, the name must be used as a singular. Therefore you should write "AM fungus inoculation", not "AM fungi inoculation". The alternative is "inoculation with AM fungi". Please, check this throughout the manuscript.
We corrected. Line 77 and 94.
This has not been corrected in the whole paper. Please do it.
5. Lines 38-39. The sentence does not work.
We corrected, line 40-41.
6. Four out of five of the isolates came from China, one from Germany. The greater similarity of geographically distant isolates was not discussed. I think it is worth doing it.
We added this point line 576-578.
> Page numbers you are referring to are not the same I have in the file (576-578 is in the bibliography!).
7. Methods. Spores were extracted throught sucrose density gradient centrifugation (lines 119-120), ok. But how were spores produced? Did you maintain plants for growing these five isolates? What kind of plants? How? This information is relevant.
We added this point line 131-139.
8. Line 140. Please rephrase.
We added this point line 143.
9. Lines 166-169. Again, the origin of the inoculum used for the experiments should be made clear, with all the necessary details!
We added this point line 131-134.
10. Coconut coir was used as substrate for plant growth. How can you state that this substrate was microrganism free (Line 159 and 182)? Was it sterilised? How?
We corrected this point line 199 and 213. The coconut coir is nature material and before import to China it is sterilized and packaged to pass customer. Afterwards although it is exposed to air naturally, but no artificial bio-effector added.
> Ok, you have not PURPOSEDLY added bioeffectors, but you do not know what an be there accidentally, after exposition to natural air!!!
11. The abbreviations of the five strains should be always the same in the text and in the figures and plots (in some cases, for instance, you use A2, in others CAUA2 - please note that E2 is indicated as CAUE, without "2" in several plots).
We corrected this.
12. Line 365. The diversity, not diverse.
We corrected.
13. Lines 376-377. Your statement is overreaching, you cannot say this, since you have not tested different levels of nutrition.
We corrected line: 416-419. The sufficient nutrient supply was according to the practical of this station used for many years with the maximum yield of tomato plants.
14. Lines 392-393. This points needs to be discussed in depth.
We discussed this point, line:458-461.
> Again, line numbers are not corresponding to what I am reading.
15. Line 410. Please, delete "while".
Corrected.
16. The comparison of the two tomato cultivars was ignored. I am not sure this is a good choice. For instance, my attention was taken by the fact that the two cultivars respond differently in terms of mineral nutrition. The autumn-winter cultivar shows nitrogen concentrations that are rather different in roots and shoots (and therefore their ratio ranges between 0.6 and 0.8), while the spring-summer cultivar has a ratio (root nitrogen concentration / shoot nitrogen concentration) that is close to 1. Something more or less opposite happens with P concentration. These effects on the nutrient accumulation might be relevant for colonization and for the efficiency of the symbiosis.
The P uptake could be related to temperature difference rather than cultivars. It cannot be compared the difference between two cultivars, due to different seasons appropriate different cultivars.
> I do not fully agree with this.It could be related to temperature differences (please provide a reference) but it could be not.
17. References. Are you sure that the name is "Azcoin" and not Azcon?
All corrcted.
> This is not exact, some corrections still remain in the text (E.g. line 64 of the pdf file).
Author Response
Thank you very much for your comments. I think it is much better in details now. We read and checked again yesterday and today.
Please see the response below, and the revised version highlighted as Track
Changes in PDF.
Review 2:
Comments and Suggestions for Authors
Manuscript plants-1293422 deals with the effects of 5 different isolates of the arbuscular mycorrhizal fungus Funneliformis mosseae on two tomato cultivars, grown in greenhouse under semi-hydroponic conditions. The topic is consistent with the journal aims and scopes. English is always understandable, although it would greatly benefit from a native speaker revision. In general the manuscript could be considered for publication, but some issues, listed below, must be addressed by the Authors.
Thank you.
Language edited by Charlesworth Author Services: Order 78861.
> I am not satisfied with this. The text still contains several mistakes. For instance (and this is not meant to be a a complete list!): the sentence on lines 15, 16 and 17 must be rephrased, "affects" should be changed in "effects" and the rest of the sentence is not understandable (numbers referring to the pdf file I was sent by the publisher - I have noticed relevant differences in page numbers you refer into your replies and this file!); "the effects on colonization rate" - the effects of what?; "positive benefits" - benefits can only be positive! (line 23); Coconut COIR (not "coin"). So, please, have this checked carefully!
All “affects”in “effects”now. thank you. Line 35.
“positive benefits”into “benefits”.line 26.
Coin into coir.
"the effects on colonization rate" “The effects of the AM fungus inoculations on” line 22,676.
- The title does not work. First, it is not correct grammatically, secondly it should be more synthetic and clear, thirdly it underlines the presence of 5 isolates (not strains) of F. mosseae, but it completely ignores that there are two plant cultivars. So, I recommend to find a better title.
We improved the title line 2-4.
> Again, the title is not satisfactory. Maybe: Beneficial effects of five different strains of...
We corrected as the new one. Beneficial effects of the five isolates of Funneliformis mosseae on the tomato plants were not related to their evolutionary distances of SSU rDNA or PT1 sequences in the nutrition solution production
- The use of the word "strain" is common in microbiology, especially with reference to bacteria. For fungi, I am convinced that the word "isolate" is more appropriate. Please, modify throughout the manuscript.
We agree your suggestion, in the case that in fig 1, the reference species used “strain”. Our opinion is to keep this word. But it is still open for further discussing.
> I stick to my suggestion to use "isolate" and not "strain". You can change figure 1, this is not a valid reason to use strain.
We changed “strain” into “isolate”.
- The abstract is very poorly written and lacking a proper organization. It makes a sudden reference to "both growing seasons" but they were not mentioned before. Please try to summarize the paper: introduction, methods, results, comment.
We improved line 21-22.
> ok.
- English: when you use a name with the function of an adjective, the name must be used as a singular. Therefore you should write "AM fungus inoculation", not "AM fungi inoculation". The alternative is "inoculation with AM fungi". Please, check this throughout the manuscript.
We corrected. Line 77 and 94.
This has not been corrected in the whole paper. Please do it.
Corrected again and checked. No “AM fungi inoculation” anymore.
- Lines 38-39. The sentence does not work.
We corrected, line 40-41. - Four out of five of the isolates came from China, one from Germany. The greater similarity of geographically distant isolates was not discussed. I think it is worth doing it.
We added this point line 576-578.
> Page numbers you are referring to are not the same I have in the file (576-578 is in the bibliography!).
Line743-746,Page 13 bottom part.
- Methods. Spores were extracted throught sucrose density gradient centrifugation (lines 119-120), ok. But how were spores produced? Did you maintain plants for growing these five isolates? What kind of plants? How? This information is relevant.
We added this point line 131-139.
- Line 140. Please rephrase.
We added this point line 143. - Lines 166-169. Again, the origin of the inoculum used for the experiments should be made clear, with all the necessary details!
We added this point line 131-134. - Coconut coir was used as substrate for plant growth. How can you state that this substrate was microrganism free (Line 159 and 182)? Was it sterilised? How?
We corrected this point line 199 and 213. The coconut coir is nature material and before import to China it is sterilized and packaged to pass customer. Afterwards although it is exposed to air naturally, but no artificial bio-effector added.
> Ok, you have not PURPOSEDLY added bioeffectors, but you do not know what an be there accidentally, after exposition to natural air!!!
We modified this sentence from “with the bio-effecter free growth media of coconut coir moss”as “with the growth media of coconut coir moss. Line 229, 243.
Due to the import production free from artificial added microorganisms are required by costumer. The coconut coir was well packaged well arriving to our research field. And we tried to avoid contamination of pathogens and other microorganisms which may effect our results.
- The abbreviations of the five strains should be always the same in the text and in the figures and plots (in some cases, for instance, you use A2, in others CAUA2 - please note that E2 is indicated as CAUE, without "2" in several plots).
We corrected this.
- Line 365. The diversity, not diverse.
We corrected. - Lines 376-377. Your statement is overreaching, you cannot say this, since you have not tested different levels of nutrition.
We corrected line: 416-419. The sufficient nutrient supply was according to the practical of this station used for many years with the maximum yield of tomato plants.
- Lines 392-393. This points needs to be discussed in depth.
We discussed this point, line:458-461.
> Again, line numbers are not corresponding to what I am reading.
Line 572-574. Page 11, at bottom of “biomass and yield production” part.
- Line 410. Please, delete "while".
Corrected. - The comparison of the two tomato cultivars was ignored. I am not sure this is a good choice. For instance, my attention was taken by the fact that the two cultivars respond differently in terms of mineral nutrition. The autumn-winter cultivar shows nitrogen concentrations that are rather different in roots and shoots (and therefore their ratio ranges between 0.6 and 0.8), while the spring-summer cultivar has a ratio (root nitrogen concentration / shoot nitrogen concentration) that is close to 1. Something more or less opposite happens with P concentration. These effects on the nutrient accumulation might be relevant for colonization and for the efficiency of the symbiosis.
The P uptake could be related to temperature difference rather than cultivars. It cannot be compared the difference between two cultivars, due to different seasons appropriate different cultivars.
> I do not fully agree with this.It could be related to temperature differences (please provide a reference) but it could be not.
- References. Are you sure that the name is "Azcoin" and not Azcon?
All corrcted.
> This is not exact, some corrections still remain in the text (E.g. line 64 of the pdf file).

Reviewer 3 Report
The authors have took into account all suggestions.
Author Response
Dear Madam/Sir,
Thank you very much for your valueable comments.
Have a nice day!
According to another reviewer suggest. the tittle improves as:
Beneficial effects of five isolations of Funneliformis mosseae on the plants were not related to their evolutionary distances of SSU rDNA or PT1 sequences in the tomato nutrition solution production
In the case your have any further comments, please do not hesitate to contact with us.
Best regards!
Zhifang
Round 2
Reviewer 1 Report
The comments and suggestions on the manuscript entitled" Beneficial efficiency of the different strains of Funneliformis mosseae on the tomato plants in the coconut coir nutrition solution production have been well addressed and improved by the authors. The change in the title is now better and satisfactorily addressing its impact.
I have further no suggestions and recommendations.
Thanks for the revised version.
This manuscript is a resubmission of an earlier submission. The following is a list of the peer review reports and author responses from that submission.
Round 1
Reviewer 1 Report
The manuscript entitled" Different beneficial efficiency of the strains of Funneliformis mosseae on the tomato plants not related to evolutionary distances of SSU rDNA or PT1 sequences in the nutrition solution production has interesting results, however, there are some minor suggestions before it proceeds for further process --
- Abstract- Line no- 17 and 18- Its hard to understand please reframe the line-
- Line no-19- Suggestions whenever authors use abbreviation they must expand it once and then abbreviations can be used in further – ex- SSu rDNA, PTI etc—
- Line no 24- its better to include the effective strain name in the paragraph.
- Introduction – well written
- Section 2.2- it was not clear from the methodology when the isolated AM fungi, how they have mass multiplied for the use in the experiment—please define and include the section.
- Line no 165 to 175,- it's confusing what authors said that the washed microorganisms--, when authors were only using Am fungi from where the microbes came – Please check and reframe—
- Growth conditions of tomato plants- authors have described other details but didn’t mention about the humidity in the greenhouse – secondly, I could also not find the soil characteristics used in the experiment—please add them if possible—
- Figures 1, 1a,b,c are not so clear if possible present a cl4easr version of the figures—
- Suggestion- in methods section Authors used stain names as the NYN1 strain, (2) A2 strain, (3) T6 strain, (4) X5 strain, and (5) tE2 strain. Then why there is different designations used in figure 2. It's hard to understand—please check
- Table 1 its unclear from table 1, if the authors used dry weight or fresh biomass. Please check and modify accordingly--
- The authors didn’t follow the format of the journal for the manuscript as well as referencing pattern. I strongly suggest going through the author's guidelines and doing the needful.
Reviewer 2 Report
Manuscript plants-1293422 deals with the effects of 5 different isolates of the arbuscular mycorrhizal fungus Funneliformis mosseae on two tomato cultivars, grown in greenhouse under semi-hydroponic conditions. The topic is consistent with the journal aims and scopes. English is always understandable, although it would greatly benefit from a native speaker revision. In general the manuscript could be considered for publication, but some issues, listed below, must be addressed by the Authors.
1. The title does not work. First, it is not correct grammatically, secondly it should be more synthetic and clear, thirdly it underlines the presence of 5 isolates (not strains) of F. mosseae, but it completely ignores that there are two plant cultivars. So, I recommend to find a better title.
2. The use of the word "strain" is common in microbiology, especially with reference to bacteria. For fungi, I am convinced that the word "isolate" is more appropriate. Please, modify throughout the manuscript.
3. The abstract is very poorly written and lacking a proper organization. It makes a sudden reference to "both growing seasons" but they were not mentioned before. Please try to summarize the paper: introduction, methods, results, comment.
4. English: when you use a name with the function of an adjective, the name must be used as a singular. Therefore you should write "AM fungus inoculation", not "AM fungi inoculation". The alternative is "inoculation with AM fungi". Please, check this throughout the manuscript.
5. Lines 38-39. The sentence does not work.
6. Four out of five of the isolates came from China, one from Germany. The greater similarity of geographically distant isolates was not discussed. I think it is worth doing it.
7. Methods. Spores were extracted throught sucrose density gradient centrifugation (lines 119-120), ok. But how were spores produced? Did you maintain plants for growing these five isolates? What kind of plants? How? This information is relevant.
8. Line 140. Please rephrase.
9. Lines 166-169. Again, the origin of the inoculum used for the experiments should be made clear, with all the necessary details!
10. Coconut coir was used as substrate for plant growth. How can you state that this substrate was microrganism free (Line 159 and 182)? Was it sterilised? How?
11. The abbreviations of the five strains should be always the same in the text and in the figures and plots (in some cases, for instance, you use A2, in others CAUA2 - please note that E2 is indicated as CAUE, without "2" in several plots).
12. Line 365. The diversity, not diverse.
13. Lines 376-377. Your statement is overreaching, you cannot say this, since you have not tested different levels of nutrition.
14. Lines 392-393. This points needs to be discussed in depth.
15. Line 410. Please, delete "while".
16. The comparison of the two tomato cultivars was ignored. I am not sure this is a good choice. For instance, my attention was taken by the fact that the two cultivars respond differently in terms of mineral nutrition. The autumn-winter cultivar shows nitrogen concentrations that are rather different in roots and shoots (and therefore their ratio ranges between 0.6 and 0.8), while the spring-summer cultivar has a ratio (root nitrogen concentration / shoot nitrogen concentration) that is close to 1. Something more or less opposite happens with P concentration. These effects on the nutrient accumulation might be relevant for colonization and for the efficiency of the symbiosis.
17. References. Are you sure that the name is "Azcoin" and not Azcon?
Reviewer 3 Report
In my opinion, and according to literature, for Arbuscular Mycorrhizal Fungi (obligate biothroph) is better the use of isolate instead of strain (generally utilized for cultivable microrganisms).
What do the authors mean by nutrition / sufficient nutrition? (Can they explain how to measure it? Are there any references for this?
Abstract:
- The term “tomato industry production with nutrient solution is to vague: please clarify starting from your abstract which type of cultivation is applied (e.g hydroponics, soilless substrate, coconut coir.…).
- Lines 18-20 “Five strains of AM fungi were collected from soils of differing geographical origins with their SSU rDNA and PTI sequences identified as Funneliformis mosseae and evidenced closing evolutionary distances”.
This sentence must be completely rewritten. The general sense is not understandable.
In particular, it must be divided into two parts: the first concerning the process of isolation and multiplication of AMF isolates, the second concerning their molecular characterization by means of two target regions.
Line 100-103 What does the author means with “the impact of a well-established mycorrhizal association”??
M&M
-It is not clear how the authors have selected and multiplying in purity the five different isolates in order to obtain sufficient inoculum to be used for the following experiments. Please add details on sub-culturing methodology to this paragraph.
Moreover, it is quite surprising to me that only Funneliformis mosseae were collected and used for the following experiments. On what assumptions did the authors based their choice? Are there any explanation for this point? How many different isolates (morphological spore types) were present in the different geographical location? Are these different location cultivated with tomatoes? This points deserve explanations and comments.
- Line 116: 18S-23S (????) rRNA gene internal transcribed spacer (ITS): It seems to me that the authors make some confusion between the bacterial intergenic spacer region (i.e prokaryotic 16S–23S rDNA intergenic spacer (ITS) and the nuclear ribosomal internal transcribed spacer (ITS) region of Fungi. Please clarify this important point.
- The authors must clarify better which primers have been finally used. It seems to me that they amplified the entire SSU and not ITS or partial 18S rDNA, then the sentence above mentioned (line 116-118) is completely wrong.
Are the 5 AMF isolates being deposited to a bank of germoplasm or herbaria? Which is their code number/ reference? Have some of these isolates ever been described and/or used elsewhere?
I wondering if is it possible to have access to the sequences obtained from the 5 AMF isolates on the 2 target regions investigated.
Line 158: It is preferable the use of “growth substrate” than “growth medium” terms when referring to solid matrices (e.g coconuts fiber, peat, sand…)
The entire paragraph AM fungi material and inoculation must be revised. It is not understandable. In this paragraph, what are the authors referring to for growth media? Coconut coir? Nutrient solution?
It is not clear when the plants were removed for the analysis. It is unclear when the plants were removed for analysis. Please provide more details on the sampling date and how many days / months have passed since inoculation.
Results
-
Figure 1 a and b: It seems to me that phylogenetic trees have been constructed without using any sequences from other Genera of AMF (e.g Rhizophagus/Rhizoglomus). I would like to see if inserting Pm1 and Pm2 sequences of of other different genera (at least just one) the topology and the bootstrap values will be the same. Moreover, reference sequences of these trees lack the accession numbers. Please insert them.
-
- Figure 1: the legend must be completely revised. Details on the phylogenetic tree construction and analyses must be added as well as accession numbers to reference sequences used.
-
In Figure 1C moreover it is not clear how the alignment on which this tree is based have been constructed (Is it a concatenated one?; What are the target region? How many characters have been analyzed for each region? Why the reference sequence are labeled as sequence
Discussion
Although were not inoculated, the root of control plants showed a low percentage of mycorrhization and the authors give an explanation for this in comparison with the inoculated ones which showed higher mycorrhization values.
I think that the first sentence of discussion (337-341) reporting this point is something quite obvious, otherwise why we do apply AMF inoculation? Instead, It is not so obvious significant mycorrhization rates under high nutient supply, as you stated some lines below. So please rephrase completely this first paragraph to be more incisive on the results obtained.
Also the lines 377-380 have to be re-written to be more clear and incisive. What the authors mean for sufficient nutrients?
Reviewer 4 Report
This paper reports a small step of understanding the different beneficial efficiency of the strains of Funneliformis mosseaeon the tomato plants not related to evolutionary distances of SSU rDNA orPT1 sequences in the nutrition solution production. However, the authors should explain what are the scientific novelties of this article compared to a previous paper on the same topic published in 2019 (Huang et al. Funneliformis mosseae Enhances Root Development and Pb Phytostabilization in Robinia pseudoacacia in Pb-Contaminated Soil ) and 2015 (Song et al. Enhanced tomato disease resistance primed by arbuscular mycorrhizal fungus) in which the same fungal pathogen were used. At the same time the fungal strains used must be correctly identified at the species and the sequences deposited in GenBank for verification. Without a clarification on these two points, the manuscript cannot be evaluated.